# Varied Bulk Powder Properties of Micro-Sized API within Size Specifications as a Result of Particle Engineering Methods

**DOI:** 10.3390/pharmaceutics14091901

**Published:** 2022-09-08

**Authors:** Zijian Wang, Marina Solomos, Stephanus Axnanda, Chienhung Chen, Margaret Figus, Luke Schenck, Changquan Calvin Sun

**Affiliations:** 1Pharmaceutical Materials Science and Engineering Laboratory, Department of Pharmaceutics, College of Pharmacy, University of Minnesota, Minneapolis, MN 55455, USA; 2Process Research & Development Merck & Co., Inc., Rahway, NJ 07065, USA; 3Analytical Research & Development Merck & Co., Inc., Rahway, NJ 07065, USA

**Keywords:** particle size, particle engineering, bulk properties, surface anisotropy, contact angle

## Abstract

Micronized particles are commonly used to improve the content uniformity (CU), dissolution performance, and bioavailability of active pharmaceutical ingredients (API). Different particle engineering routes have been developed to prepare micron-sized API in a specific size range to deliver desirable biopharmaceutical performance. However, such API particles still risk varying bulk powder properties critical to successful manufacturing of quality drug products due to different particle shapes, size distribution, and surface energetics, arising from the anisotropy of API crystals. In this work, we systematically investigated key bulk properties of 10 different batches of Odanacatib prepared through either jet milling or fast precipitation, all of which meet the particle size specification established to ensure equivalent biopharmaceutical performance. However, they exhibited significantly different powder properties, solid-state properties, dissolution, and tablet CU. Among the 10 batches, a directly precipitated sample exhibited overall best performance, considering tabletability, dissolution, and CU. This work highlights the measurable impact of processing route on API properties and the importance of selecting a suitable processing route for preparing fine particles with optimal properties and performance.

## 1. Introduction

Particle size plays a crucial role in the performance of pharmaceutical products. For example, the dissolution behavior of an active pharmaceutical ingredient (API) depends on particle size, where larger specific surface area of smaller API particles yields faster dissolution according to the theory of the Noyes-Whitney equation [1]. Smaller API particles in the sub-micron range may also exhibit higher solubility [2]. Therefore, micron-sized or nano-sized API particles have been commonly employed to improve bioavailability of poorly soluble APIs [3]. Tabletability is also affected by particle size, where a larger surface area available for bonding between particles in a tablet contributes to higher tabletability [4]. Conversely, larger particle size favors flowability [5]. Blend and tablet content uniformity (CU) may be improved from a purely statistical standpoint by micronizing API [6,7], but CU is also affected by segregation or agglomeration of API particles during manufacturing [8]. For poorly soluble APIs, e.g., biopharmaceutical classification system (BCS) class II and IV APIs, it is imperative to enhance their dissolution and bioavailability [9]. To this end, several particle engineering techniques have been used to prepare micron-sized APIs in the pharmaceutical industry [10]. Among them, milling is representative of the “top-down” methods, where large particles are reduced into micron-sized particles through breakage by mechanical impact, and precipitation is representative of the “bottom-up” methods, where API molecules in solution assemble into micron-sized particles.

Milling can be performed either dry or in a liquid medium. Compared to dry milling, wet milling (e.g., media milling or high pressure homogenization) usually produces smaller and smoother particles with a lower tendency to agglomerate, though filtering and isolating wet-milled particles can be difficult [10,11]. Among available dry milling processes, air-jet milling is preferred in pharmaceutical industry over ball milling [12], pin milling [13], and hammer milling [14], due to its high efficiency and the absence of moving parts. During jet milling, fast air flow moves particles and causes collision, attrition, shear, and compression to reduce particle size [15]. The high energy input of this method may result in a disordered crystal lattice, potentially leading to the formation of amorphous content, or polymorphic transition [10,16,17,18]. There is also the risk of product contamination by fine metal particles shed during milling.

In the classic “bottom-up” precipitation method, a high concentration API solution in a solvent is mixed with a miscible anti-solvent, resulting in high supersaturation and fast precipitation of fine particles. Particle size of precipitated API can be controlled by changing crystallization parameters and solvent systems [19]. However, rapidly formed small crystals also tend to aggregate, resulting in a broad particle size distribution (PSD). In addition to size, crystal morphology may be affected by experimental conditions and solvent system, resulting in different surface area, dissolution, flowability, and tabletability of an API even when a similar PSD is maintained [20,21,22,23]. Moreover, different faces of molecular crystals may have different surface properties due to the presence of different functional groups. Such surface anisotropy may lead to different wettability and dissolution behavior of the same API solid form prepared through different routes [18,20,24,25,26].

This work sought to systematically investigate how different particle engineering routes, while reaching equivalent biopharmaceutical performance of an API, may impact key bulk powder properties. Such knowledge informs the selection of a route to meet the target particle size specifications while optimizing powder properties. Odanacatib, an inhibitor of cathepsin K [27], was selected as a model API. The particle size specifications <6 µm (M_v_) was established through semi-mechanistic pharmacokinetic/pharmacodynamic (PK/PD) models and *in vivo* data to demonstrate bioequivalence criteria based on AUC and *C*_max_. Jet milling and liquid anti-solvent precipitation approaches were selected to prepare 10 batches of fine API with particle size in the target range. In addition, jet milling with processing aids to reduce milling induced disorder was assessed for possible impact on dissolution. 

## 2. Materials and Methods

### 2.1. Materials

Various lots of Odanacatib were obtained from Merck & Co., Inc. (Rahway, NJ, USA). Ethanol (200 proof) was purchased from Decon Labs, Inc. (King of Prussia, PA, USA). DMF and acetone were used to prepare precipitated samples and were purchased from Fisher Scientific (Fair Lawn, NJ, USA). Pharmaceutical excipients used in this work, i.e., Avicel PH102 (MCC; FMC Biopolymers; Newark, DE, USA), spray-dried lactose monohydrate (LM; Foremost; Baraboo, WI, USA), Kollidon VA64 (Crospovidone; BASF; Ludwigshafen, Germany), sodium stearyl fumarate (SSF; JRS Pharma; Patterson, NY, USA) magnesium stearate (MgSt; Covidien, Dublin, Ireland) were used as received.

### 2.2. Methods

#### 2.2.1. Preparation of Samples

A bulk sample of Odanacatib, referred to as Sample A, was re-crystallized from an acetone and water mixed solvent system. Odanacatib was suspended in 8 volumes of a 1:2 acetone–water mixed solvent. The slurry was agitated with mixing to just suspend solids and heated to 45 °C over 60 min to partially dissolve it and held at 45 °C for 15 min before cooling to 20 °C over 10 h. This heating–cooling cycle was repeated a second time. Solids were filtered, washed with water, and dried.

Sample A was separately jet milled at 300 kg scale to create Samples A1 and A2. Samples A3, A(S) and A(M) were all milled starting from Sample A using a spiral mill (Jet Pulverizer 2” Micron Master). Sample A3 was generated by milling 100 g of Sample A with an injection pressure of 100–120 psi, and a grinding pressure of 50 psi. Sample A(S) was first prepared by blending 99 g of Sample A with 1 g of sodium stearyl fumarate (SSF) on a LabRam acoustic mixer at 60 Hz for 10 min, generating a 1 wt% coated SSF sample. The blend was then milled using conditions scaled down from milling Samples A1 and A2, with an injection pressure of 100–120 psi, and a grinding pressure of 50 psi. Similarly, Sample A(M) was prepared by first blending 99 g of Sample A with 1 g of magnesium stearate (MgSt) on a LabRam acoustic mixer at 60 Hz for 10 min, generating a 1 wt% coated MgSt sample. The blend was then milled with an injection pressure of 100–120 psi, and a grinding pressure of 50 psi. Processing aids, MgSt (hydrophobic) and SSF (hydrophilic), were used to alleviate possible disorders introduced by milling. To remove amorphous content, A2 was annealed at 40 °C and 75% relative humidity for 3 weeks (Sample A2 (3 wks)) and 5 weeks (Sample A2 (5 wks)).

Sample P1 and Sample P2 were prepared from high shear direct precipitation. A rotor/stator wet mill was used to achieve a high shear environment for precipitation. The following parameters were used: (1) Quadro HV0 with 6 mm/3 mm emulsion tooling operating at 70 m/s; (2) Solvent to antisolvent volume ratios for precipitation were consistently 1:10, with addition of solvent to the mill head at an approximate flow rate 10% of the flow rate of the antisolvent. Sample P1 was prepared by dissolving Sample A in 2.5 volumes of DMF and precipitating in 25 volumes deionized (DI) water pre-cooled to 0.2 °C and seeded with 1 wt% of A1. Precipitation was completed over 90 s using a tip speed of 70 m/s. The slurry was aged for 72 h at room temperature before filtration, followed by washing with antisolvent, and drying using a nitrogen sweep applied across a filter funnel until ICH residual solvent specifications were met. Sample P2 was prepared by dissolving Sample A in 3.85 volumes of a 9:1 acetone-water mixed solvent system and precipitating in 38.5 volumes of DI water pre-cooled to 0.2 °C and seeded with 1 wt% of A1. Precipitation was completed over 90 s using a tip speed of 70 m/s. The slurry was aged for 4 h at 50 °C before being filtered, washed with antisolvent, and dried.

To investigate effects of starting API morphology on properties of milled Odanacatib, Sample B was obtained from recrystallization in acetone and methyl tert-butyl ether (MTBE). Odanacatib was suspended in 20 volumes of 3:2 acetone–MTBE mixed solvent system and heated to 50 °C to dissolve all solids. The batch was held at 50 °C for 10 min, then cooled to 37 °C over 15 min, and 2 wt% Sample A1 was added as seeds. The seeded batch was aged for 60 min at 37 °C, then 16 volumes of MTBE was dosed over 5 h. The batch was cooled to 20 °C over 10 h and subsequently filtered and dried to generate Sample B. Sample B1 was generated by milling 100 g of Sample B with an injection pressure of 100–120 psi, and a grinding pressure of 50 psi. These samples are summarized in Figure 1 for easy comparison. 

#### 2.2.2. Particle Size Distribution

Particle size distribution (PSD) measurements were made using a laser diffraction particle size analyzer (Microtrac S3500, York, PA, USA) having a detector system located at a distance from the point where the particles interact with the light. The laser light (λ = 780 nm) allows for measurement of particles by detecting the light scattered over an angular range of 0.02° to approximately 45° angle. The instrument was set for measuring irregular solid API particles, using a refractive index of 1.51. The circulating media was Isopar-G with refractive index of 1.42. Volume distribution of particles was used in reported particle size data. For each measurement, the sample was sonicated at 30 W for 120 s to disperse particle aggregates.

#### 2.2.3. X-ray Diffractometry

Powder X-ray diffractometry (XRD) and tablet XRD were collected on a powder X-ray diffractometer (PANalytical X’pert pro, Westborough, MA, USA), using Cu Kα radiation (1.54056 Å). Samples were scanned with a step size of 0.02° and 1 s/step dwell time from 5° to 35° 2*θ*. The tube voltage and amperage were 45 kV and 40 mA, respectively. Tablets for XRD were prepared at a compaction pressure of 400 MPa for 1 min on a material testing machine (model 1485; Zwick/Roell, Ulm, Germany).

#### 2.2.4. Scanning Electron Microscopy (SEM)

Samples were sputter-coated with gold using a sputter coater (Electron Microscopy Service Q150R, Hatfield, PA, USA) and images were taken using a scanning electron microscope (Hitachi SU-3400, Dallas, TX, USA). Each image was obtained using the secondary electron detector with 2 keV accelerating voltage under high vacuum.

#### 2.2.5. Surface Area Analysis

Specific surface area of each API lot was obtained from analyzing low-temperature nitrogen adsorption-desorption isotherms (at 77 K) collected using a TriStar II analyzer (Micromeritics Instrument Corp., Norcross, GA, USA). Each material was loaded into a sample tube and degassed under nitrogen at 35 °C for 1 h before analysis. After cooling to room temperature, the tube was weighed and placed into the adsorption port of the instrument. A static adsorption mode was used including full equilibration after each adsorbate load. The adsorption isotherms were measured over a relative pressure, p/p_o_, range of 0.001–0.995. Desorption isotherms were measured over a relative pressure range of 0.995–0.015. Surface area was calculated via the Brunauer–Emmett–Teller (BET) method using the relative pressure range from 0.1 to 0.30 [28].

#### 2.2.6. Tabletability

Tabletability is the capacity of a powder to be transformed into a tablet of specified strength under the effect of compaction pressure [29]. An API powder with poor tabletability cannot be made into sufficiently strong tablets by compaction. Thus, an appropriate formulation must be developed to enable successful manufacturing of tablets. A compaction simulator (Styl’One, Medelpharm, Beynost, France) was used to prepare tablets for the powder tabletability study. Forces on the upper and lower punches were captured with force sensors (strain gauges), while punch displacements were monitored using incremental sensors. Round (8 mm diameter) flat-faced punches were used for all compaction experiments. For each powder, a series of compacts were obtained in the compaction pressure range of 20 to 350 MPa. All compaction experiments were performed at a speed corresponding to a 103 ms dwell time. MgSt spray (Styl’One Mist) was used to lubricate the die wall and punch tips.

All tablets were relaxed for at least 24 h before measuring their diameters and thicknesses using a digital caliper. Diametrical breaking force was determined using a texture analyzer (TA-XT2i, Texture Technologies Corp., Scarsdale, NY, USA) at a speed of 0.001 mm/s with a 5 g trigger force. Tablet tensile strength was calculated from the maximum breaking force and tablet dimensions using Equation (1).
(1)$$\sigma =\frac{2F}{\pi DT}$$
where *F* is breaking force, *D* is diameter, and *T* is thickness.

#### 2.2.7. Shear Cell Testing

A ring shear cell tester (RST-XS, Dietmar Schulze, Wolfenbüttel, Germany), with a 10 mL cell, was used to perform powder flow testing (*n* = 3) at pre-shear normal stress of 3 kPa by following a standard 230 method [30,31]. To perform a shear test, a shear cell was overfilled with a powder of interest and excess powder was gently scraped off using a spatula to obtain a surface flush with the upper edge of the shear cell. Care was taken to avoid compression or shaking of the powder bed.

#### 2.2.8. Intrinsic Dissolution Rate

Intrinsic dissolution rate (IDR) was measured using a rotating disc method [32]. Each powder was compressed at a pressure of 400 MPa for 1 min by a material testing machine (model 1485; Zwick/Roell, Ulm, Germany) with a custom-made stainless-steel die, against a flat stainless steel disc for 2 min to prepare a pellet (6.39 mm in diameter). A range of compaction pressures were assessed, but this pressure was necessary to avoid shedding of micronized particles from the faces of the compacts, which would convolute the dissolution performance. The obtained pellet had a visually smooth surface that was coplanar with the surface of the die. While rotating at 300 rpm, the die was immersed in 300 mL of pure ethanol in a water-jacketed beaker. An UV-vis fiber optic probe (Ocean Optics, Dunedin, FL, USA) was used to continuously monitor the UV absorbance of the solution at 267 nm, which was converted to a concentration–time profile based on a previously constructed concentration–absorbance standard curve. The initial linear part of the dissolution curve was used for calculating the dissolution rate.

#### 2.2.9. Contact Angle

Compacts of each sample were prepared by a Styl’One compaction simulator at 300 MPa to attain an essentially pore-free surface for measuring contact angle by the sessile drop method using a goniometer (DM-CE1; Kyowa Interface Science Co. Ltd., Niiza, Japan). A drop of 2 μL of DI water was placed on the surface using a syringe dispenser. The equilibrium contact angle, *θ*, between the sample surface and the tangent line at the edge of the drop was determined from the captured images using the software (FAMAS, Kyowa Interface Science, Niiza, Japan). Three measurements made on different tablets of each API sample were used to calculate the mean and standard deviation.

#### 2.2.10. Content Uniformity (CU)

Samples A, A1, B1, P1 and P2 were selected to study the CU of a generic formulation consisting of 1% API, 27% MCC, 66% lactose monohydrate, 5% crospovidon, and 1% MgSt. To maximize uniformity, blends (50 g batch size) were prepared by layering ingredients in a bottle in the following order: LM, API, crospovidon, MCC. The powder was then mixed on a shaker mixer (Turbula T2F, Glen Mills Inc., Clifton, NJ, USA) for 10 min at 49 rpm. Then, MgSt was added to the blend and further mixed for 2 min. A total of 100 tablets (300 mg tablet weight) of each formulation were prepared at a compaction pressure of 200 MPa on Styl’One, and 10 out of the 100 tablets were randomly selected to determine API amount for assessing CU measurement by following USP <905> [33].

Each tablet was weighed and ground into a powder. Subsequently, the powder was transferred into 100 mL of pure ethanol to dissolve API and passed through a syringe filter with a 0.45 µm membrane. Then, 2 mL of the filtered solution was diluted to 10 mL with ethanol. The concentration of the resulting solution was determined from absorbance at 267 nm (UV-vis fiber optic probe) using a previously constructed calibration curve.

#### 2.2.11. Statistical Analysis

To assess statistical significance of difference, the one-way analysis of variance (ANOVA) and Tukey’s multiple comparisons test were performed for all API samples, at a *p* < 0.05 level [34].

## 3. Results and Discussion

### 3.1. Solid-State Characterization

#### 3.1.1. Particle Size Distribution and Specific Surface Area

The two starting API lots for jet milling, Samples A and B, were much larger in particle size than all milled samples (Table 1), suggesting the effectiveness of particle size reduction by jet milling. The PSDs of the three milled API lots from A are similar (Table 1). Crystals in Sample B1 are clearly smaller than those of A1, A2, and A3 lots. The use of processing aids led to smaller particles in both A(S) and A(M) than those processed without using a processing aid. It is possible that lubricated API particles agglomerated to a lesser extent and could travel with the nitrogen jet at a higher speed, leading to a more effective size reduction due to a higher energy of impact. The Sample P2 contained particles larger than those in Sample P1, indicating a clear impact of desaturation kinetics on the precipitation process. Importantly, the particle sizes of all engineered API samples (Table 1) meet the size specifications, i.e., *M*_v_ < 6 µm, which was deemed sufficient for equivalent clinical biopharmaceutical performance of Odanacatib. Thus, despite their different mechanisms, both jet milling and fast precipitation could generate fine particles meeting the size specifications. This opens an opportunity for investigating potentially different powder properties of such biopharmaceutically equivalent API lots, and their impact on tablet CU.

As expected, all milled API samples had larger specific surface area (SSA) than their respective parent API (Table 1). Sample A1 had smaller SSA than A2 and A3, which is consistent with the overall smaller *M*_v_ of A1 (Table 1). Sample A(S) had significantly larger SSA than A(M). This was attributed to the overall smaller particles in A(S). The precipitated Sample P2 had SSA about 50% of that of P1, which is consistent with the overall larger particles in Sample P2 (Table 1). Among the milled samples, A1 had a broader PSD as suggested by its larger span. Similarly, among the two precipitated samples, P1 exhibited a broader PSD than P2 (Table 1).

#### 3.1.2. Crystal Morphology

SEM images of all samples were obtained to compare their crystal morphologies. Crystals in the unmilled Sample A were rod-like while crystals in Sample B were needle-like, with a much higher aspect ratio than Sample A (Figure 1). All engineered samples, however, had a comparable crystal morphology.

### 3.2. Crystallinity and Bulk Powder Properties 

The PXRD patterns of all engineered samples were consistent with the calculated pattern from single crystal data. However, the parent API lots, A and B, exhibited visible differences in the range of 17 to 21 degrees (Figure 2a), which may be attributed to the preferred orientation effect due to their elongated crystal morphologies. This hypothesis gained support from the observation that milled lots of both materials exhibited similar XRD patterns that conform to the calculated XRD pattern (Figure 2a), suggesting that all milled and precipitated samples were the same solid form.

As all the XRD data were normalized to minimize the possible impact of different sample size and variations in X-ray experimental parameters, the degree of crystallinity of samples is better assessed by peak width than peak intensity [35,36]. The tablet XRD of all samples showed clearly broader peaks than corresponding powder XRD patterns (Figure 2b), suggesting inherent sensitivity of Odanacatib crystals to mechanical stresses. This is consistent with the observation that peaks are sharper in the two unmilled API lots, A and B, than peaks in milled samples (Figure 2a). Among milled samples, A2 exhibited broadest peaks with the poorest resolution of neighboring peaks (Figure 2a). All these observations establish the sensitivity of Odanacatib crystals to external mechanical stresses, which are encountered during both milling and compaction.

The tabletability plots of all samples are markedly different (Figure 2c). Sample A, with larger crystals, exhibited better tabletability than all of its corresponding milled samples, which could not form intact tablets due to severe lamination. We speculate that air entrapment could have caused the different tableting behaviors, since milled API materials are more cohesive and it is more difficult for air to escape the die during the course of compression. Consequently, the expansion of entrapped air during decompression led to lamination of tablets [37,38]. This hypothesis is supported by the observation that intact tablets of all samples could be obtained using a slow tableting speed when preparing specimens for IDR and contact angle measurements. The air entrapment mechanism is also consistent with the fact that it is more difficult for air to escape from the cotton-ball-like Sample B, which consisted of long entangled needles and laminated upon tablet ejection. The better tabletability of P2 powder compared to Sample A can be explained by the fact that, in the absence of lamination by entrapped air, the smaller size of P2 led to stronger tablets than A due to a larger bonding surface area of P2.

All micro-sized API samples were very cohesive based on their flowability index (FFC) values (Figure 2d) [39], which is characteristic of fine particles. With a larger particle size, unmilled Sample A exhibited a better flowability than all corresponding milled samples (Figure 2d), which is expected [40]. The flowability of Sample B could not be measured by shear cell, because it consisted of large cotton-ball like agglomerates of crystals, and a flat sample surface could not be prepared.

The significantly different tabletability and flowability (*p* < 0.0001) of the investigated API powders could be attributed, in part, to the slightly different particle sizes (Table 1). However, the different surface energies arising from surface anisotropy is also expected to play an important role. To confirm the possible differences in surface energy, we studied their IDR as this property is independent of the particle size of the API. We also directly assessed differences in surface energy by contact angle measurement. 

Because of the low aqueous solubility of Odanacatib, our attempts to measure IDR in aqueous media was unsuccessful due to poor precision. Hence, pure ethanol was used as a dissolution medium for IDR testing. However, ethanol could not be used for contact angle measurement due to difficulty with delivering a drop onto the sample surface. As a result, we used DI water to probe wettability. Significant differences among the IDR (Figure 2e, *p* < 0.0001) and the contact angle (Figure 2f, *p* = 0.0009) data of all engineered samples were observed, suggesting their different surface energetics. Thus, differential surface energy is a factor when explaining the different bulk properties of these API samples. To gain a better understanding of how the preparation routes affected the bulk properties of API powders, we have systematically examined effects of various process variables in a pair-wise fashion in the following section.

### 3.3. Comparison of Various Process Variables

#### 3.3.1. Inherent Process Variability of Jet Milling (A1 vs. A2) 

A1 and A2 were different batches prepared by jet milling Sample A at a commercial manufacturing scale under the same set of processing conditions. However, A2 has significantly lower crystallinity than A1, as suggested by broader peaks in the powder XRD pattern of A2 (Figure 3a). For example, adjacent peaks at ~15, 17.5 and 20 degrees could barely be distinguished in the PXRD pattern of A2 due to peak broadening, but these peaks were clearly resolved in the PXRD pattern of A1. The different crystallinities could have been caused by (1) a higher amorphous content generated in Sample A2 during jet milling, and (2) more of the amorphous content in Sample A1 crystallized during the storage. Since both samples had been stored at ambient conditions for several years, it is more likely that the first factor was responsible for the different crystallinity, i.e., external stresses applied onto crystals were somehow lower when preparing A1 by milling. This is supported by the observation that differences in crystallinity between A1 and A2 were diminished after compression at 400 MPa (Figure 3a), i.e., A2 tablet only showed a slightly lower crystallinity than that of A1. This can be explained by the compaction pressure being higher than the stresses during milling. Hence, the crystallinity of the tablets was more affected by compaction, which was the same for the two samples, rather than milling. This is consistent with the observation that their IDRs are not significantly different (Figure 3c), despite the higher amorphous content in A2 powder. The water contact angle of A1 tablet is also not significantly different from A2 (Figure 3d). A1 exhibited significantly better flowability than A2 (Figure 3b, *p* = 0.0003). However, differences in the tabletability of A1 and A2 could not be demonstrated, because neither powder could form intact tablets under high-speed compression. The differences in amorphous content between A1 and A2 highlight inherent variability in a full-scale jet milling process.

Thus, the seemingly same set of jet milling parameters did not guarantee the same properties of milled API, suggesting potential milling parameters not robustly controlled, or perhaps storage conditions of the two samples in time following milling were different.

#### 3.3.2. Effects of Starting Material on Jet-Milled API (A3 vs. B1)

To assess the impact of starting materials on the milled API, two batches of morphologically different Odanacatib (A and B) were jet milled at laboratory scale under identical milling conditions to obtain A3 and B1. The control of milling parameters at the laboratory scale was expected to be robust. Hence, we assumed that the differences between A3 and B1 were mostly due to the differences in the starting materials. However, contributions from process variations cannot be excluded. Under given milling conditions, the fracture of crystals depends on the presence of energetically weak cleavage planes, size, and aspect ratio. Fracture along cleavage planes is energetically favored for crystals in A because of the relatively lower aspect ratio. However, breaking crystals along the long dimension for B crystals is statistically preferred. If such fracture surfaces are not the same as the cleavage plane, the distributions of exposed crystal surfaces in the two milled samples could be different even if their milled crystals have closely similar size and shape (Figure 1 and Table 1).

Although powder XRD patterns of A3 and B1 are comparable, the XRD peaks of the B1 tablet in the 17–20 degrees region were broader than those of A3 tablet (Figure 4a and Appendix A), suggesting that the crystals in Sample B1 were more prone to stress-induced disorders during compaction. The flowability of A3 was significantly worse than B1 (Figure 4b, *p* = 0.0002). Since crystals in A3 were larger than those in B1 (Table 1), difference in flowability cannot be explained by differences in size. Hence, we attribute it to different surface energies of these milled samples. Compared to B1, A3 showed a higher IDR (Figure 4c) and a larger contact angle by water (Figure 4d), but the differences were not statistically significant. This suggests that differences in surface energies, which led to different flow properties between the two API lots, were largely reduced by compaction. 

#### 3.3.3. Effects of the Milling Scale (A1 vs. A3)

Successful scale-up is a challenge in the development of drug products [41,42]. Potential effects of the scale of jet milling were assessed in this study by comparing A1 to A3, which were generated from the same starting material at an industrial scale and a laboratory scale, respectively.

The crystallinity, IDR, and contact angle of A1 and A3 did not significantly differ (Figure 5a,c,d). However, A1 exhibited a statistically better flowability than A3 (Figure 5b, *p* = 0.0002), despite the smaller sizes of crystals in A1 (Table 1). This observation suggests that materials generated by jet milling at different scale can exhibit different powder properties, likely due to different external stresses during milling, leading to differences in exposed surfaces in the milled materials. Overall, the impact of scale effects are less than the effects caused by different starting materials discussed in the preceding section.

#### 3.3.4. Jet-Milled vs. Precipitated Samples (A3 vs. P2)

To assess the extent of the impact of particle engineering techniques on bulk properties of micro-sized particles, we compared jet-milled Sample A3 to precipitated P2. Jet milling produces fine particles from large ones by mechanical stresses (top-down). Fast precipitation produces fine particles from the solution through fast nucleation and growth (bottom-up). Given the significantly different preparation mechanisms, different crystal surfaces are generated, exhibiting different surface energetics and powder properties. We chose P2, instead of P1, for the pair-wise comparison, because its PSD is closer to that of A3 (identical *d*_50_ and overlapping PSD, Table 1).

Both powder and tablet XRD patterns of the two samples revealed no detectable differences (Figure 6a). However, P2 could form intact and strong tablets in the entire pressure range investigated but no intact tablet of A3 could be obtained (Figure 6b). The difference in tabletability was attributed to their different surface properties, since they exhibit comparable crystallinity and PSD (Table 1). In fact, crystals in P2 appear to have smoother surfaces than those in A3. Rougher particle surfaces generally lead to more difficult air escape from a powder bed during compaction. This explains why intact tablets could not be prepared from A3 under fast compression but could be prepared under a slower compression speed when preparing pellets for contact angle measurements. The rougher crystal surfaces in A3 also explain its significantly worse flowability than P2 (Figure 6c, *p* = 0.0216). The IDRs and contact angles of the two samples differed but the differences are not statistically significant (Figure 6d,e). Again, any differences in surface properties would have been minimized by the process of compaction, which amorphized both powders to about the same extent (Figure 6a).

#### 3.3.5. Effects of Solvent Systems in the Precipitation Route (P1 vs. P2)

The properties of an API powder from a fast precipitation process, such as crystallinity, particle size and particle morphologies, are known to be affected by process factors such as ratio of solvent to anti-solvent, mixing rate, desaturation kinetics, and type of solvent/anti-solvent pair [10]. The selection of the solvent systems is typically based on three factors, i.e., miscibility between the pair of solvents, viscosity ratio between the solvent pair (less than 3), and a solubility phase diagram that favors anti-solvent precipitation [43]. Thus, we have evaluated effects of solvent systems on material properties using DMF/water (P1) and acetone/water (P2) solvent pairs.

Powder XRD patterns revealed no major differences between the two samples (Figure 7a), suggesting negligible impact of solvent type on crystallinity of precipitated API. However, crystallinity of P1 tablet is lower than that of P2 tablet, as suggested by the broader diffraction peaks of P1 tablet (Appendix A). P2 exhibited significantly better tabletability than P1 (Figure 7b), which can be attributed to the smaller sizes of crystals in P1 (Table 1, Figure 1). For the same material, smaller particles lead to more cohesive powders, exhibiting higher porosity. Thus, there was more air in the P1 powder that needed to escape during compaction. Additionally, air escape from a powder bed consisting of smaller particles is also less efficient [37]. Taken together, severe lamination of P1 tablets was observed when compressed at a high speed, where little time was available for air to escape the powder bed. Similar to the tableting behavior of A3, P1 powder could form intact tablets when compressed at a slower speed to form pellets for contact angle experiments. 

The smaller particle size of P1 was expected to cause poor flowability. However, no significant difference was observed (Figure 7c). This was likely because any differences are masked by the slightly large variability in the measured flowability of P2. P2 exhibited faster dissolution rate than P1 (Figure 7d, *p* = 0.0002), indicating more favorable surface characteristics for wetting by ethanol. This was because tablet XRD suggested lower crystallinity of the P2 tablet, which favors faster dissolution because of the higher free energy of a more disordered solid. However, both samples did not show a statistically significant difference of contact angle with water (Figure 7e). It is likely that any difference in contact angle was masked by the relatively high degree of variability in the measured contact angle value of P1. 

#### 3.3.6. Effects of Crystallinity

The previous pairwise comparison could not rule out potential effects of different particle sizes on powder properties. The less crystalline Sample A2 provides an opportunity to study the impact of crystallinity on powder properties without significant impact from different particle sizes. This was done by annealing A2 at 40 °C and 75% relative humidity for different lengths of time to obtain API with increasing degrees of crystallinities through inducing crystallization of amorphous domains [44,45]. Longer annealing time led to higher crystallinity (Figure 8a), as expected. Upon compression, a more crystalline powder also formed tablets with higher crystallinity (Figure 8a).

The results show that higher crystallinity corresponded to better flowability (Figure 8b), which is consistent with the earlier observation that more crystalline A1 exhibited significantly better flowability than A2 (Figure 3d). Although both IDR and contact angle are expected to differ among the three samples of different degrees of crystallinity, observed differences were not statistically significant (Figure 8c,d), which may be a result of compression induced amorphization.

#### 3.3.7. Effects of Processing Aids during Jet Milling (A3 vs. A(S) vs. A(M))

Given the observed sensitivity of Odanacatib to external stresses, processing aids with known lubricating functionality were used to reduce the amorphous content in jet-milled API. A(S) was jet-milled Sample A with 1% SSF, a hydrophilic lubricant, and Sample A(M) was prepared by milling Sample A with 1% MgSt, a hydrophobic lubricant. A jet-milled lot without any processing aid, A3, was used as a reference sample to evaluate the impact of processing aid on bulk properties.

All three samples were highly crystalline based on their powder XRD patterns. Based on peaks within the range of 17 to 21 degrees, crystallinity of A(M) and A(S) were both higher than A3 (Appendix A). Thus, the use of processing aids did alleviate milling induced amorphization of Odanacatib. We note that diffraction peaks at 2Θ below 16 degrees were unexpectedly much less intense for A(S) than A(M) (Figure 9a). Although a clear explanation for this observation is elusive, preferred orientation is unlikely due to the small crystal sizes in both samples and the relatively low aspect ratios of crystals (Figure 1). However, tablets of all three powders exhibited significantly lower crystallinity, again confirming the stress sensitivity of Odanacatib crystals (Figure 9a).

A(S) showed significantly better flowability than the other two samples (Figure 9b), despite its smaller particle size (Table 1). Thus, the presence of SSF not only reduced amorphization but also improved powder flowability. However, the use of MgSt as a processing aid did not improve flowability as reported for other materials [46]. The use of a processing aid; however, significantly reduced the IDR (Figure 9c). The extent of reduction was more significant by the hydrophobic lubricant, MgSt, which is expected because wetting would be more difficult with MgSt present on the tablet surface. In addition, the coverage of API particle surfaces by lubricant particles would also have led to slower dissolution. Wettability by water followed a descending order of A(S) > A3 > A(M) (Figure 9d). This is expected as SSF is hydrophilic, but MgSt is hydrophobic. 

The result indicated that, when a processing aid is needed to reduce the extent of amorphization of API by milling, a hydrophilic processing aid is preferred.

### 3.4. Impact on Content Uniformity in Tablets

So far, it is clear that Odanacatib batches using different particle engineering routes, while meeting size specifications for biopharmaceutical performance, exhibited different powder properties and processability. If the API loading is high, such differences are expected to impact corresponding properties of final blends. When the API loading is low, such differences may influence tablet CU. The latter effect was investigated using tablet formulations with 1% API loading. Since CU is typically sensitive to particle size but the PSDs of these API samples varied, two samples from each particle engineering technique covering a wide range of PSD were studied. Sample A with larger particle size was also selected as a control.

Among all five formulations prepared, the formulation of Sample A surprisingly exhibited the best CU (Figure 10a). At a constant 1% API loading, from a purely statistical perspective, smaller API particles should lead to better CU, provided API particles are uniformly distributed in the blend. This counterintuitive observation was attributed to the lowest agglomeration tendency of the A formulation during storage. Micro-sized API formed large agglomerates during storage to different extents (Figure 10b). In general, higher agglomeration tendency correlates with poorer flowability (Figure 2d). 

Among the micro-sized API batches, CU of tablets containing API generated by milling (A1 and B1) was poorer than that of those by precipitation (P1 and P2) (Figure 10a). This was likely due to the fact that, compared to precipitation, the milling process led to API crystals with higher surface energy, and subsequently a higher agglomeration tendency. Compared to A1 and B1, P1 and P2 agglomerates were also weaker and could be broken without applying a large force. Thus, agglomerates observed in Sample P2 could have been broken during powder feeding and mixing immediately before compression.

Based on all results in this study, API lots prepared through different engineering routes exhibited significantly different powder properties, which impact performance when used in a tablet formulation. Among these Odanacatib batches, P2 has the overall best powder properties and performance for tablet manufacturing.

## 4. Conclusions

This study showed that, while different engineering routes can be used to produce Odanacatib lots that meet the target Mv, they do have a measurable impact on a range of pharmaceutical properties, including solid-state properties, powder properties, dissolution, and content uniformity. While many more subtle trends can be deduced from the data set, general avoidance of fines seems a key consideration. Samples with higher *d*_10_ (A3, P2) showed fastest IDR, reasonable compression performance and flow, while the samples with the lowest *d*_10_ showed poor flow and slowest IDR (not accounting for samples milled with lubricants). While the addition of processing aids may reduce disorder during milling, this had a detrimental impact on IDR, and did not appreciably improve flow. Hence, beyond API size specifications based on biopharmaceutical considerations, properties influencing robust manufacturability and drug product performance should also be considered when selecting a particle engineering route for drug substance manufacturing. For Odanacatib, the direct precipitated Sample P2 exhibited overall best performance (tabletability, dissolution, and content uniformity) relative to the other processed samples. The new insights attained from this work help the selection of an optimal processing route for preparing micronized Odanacatib. This systematic methodology for evaluating the performance of a diverse range of engineered particles can be applied to other active pharmaceutical ingredients to ensure a globally optimum process and product.

## Data Availability

The data presented in this study are available upon request.

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
