# Peer review of "Varied Bulk Powder Properties of Micro-Sized API within Size Specifications as a Result of Particle Engineering Methods"

_pharmaceutics, 2022, doi:10.3390/pharmaceutics14091901_

Round 1

Reviewer 1 Report

The manuscript entitled "Varied Bulk Powder Properties of Micro Sized API within Size Specifications as A Result of Particle Engineering Methods " is a study of the bulk properties’ variability in some different batches of Odanacatib prepared through either jet milling or fast precipitation. The work is interesting as the particle size reduction is one of the several formulation strategies proposed for the improvement of the poorly water-soluble APIs. Although a significant number of experiments was conducted, there are still many concerns that need to be clarified in order for this paper to be considered as acceptable.

1.       The manuscript, and specifically the abstract, needs a thorough check of English language.

2.       The writing and the logical order of the paper’s Introduction needs to be reorganized. For example, the authors should reduce the size of the paragraph at lines 30-44, since it essentially constitutes a repetition of the same statement. Furthermore, a brief explanation of the “top-down” and “bottom-up” methods should be provided.

3.       Tabletability is a critical attribute of the examined samples which is evaluated in detail at this study. That being said, the authors should provide the definition of the term and highlight its importance.

4.       The definition of the abbreviations when they first appear in the manuscript is mandatory. The authors should thoroughly check the manuscript in order to define all the abbreviations used. For example, BCS at line 42, MTBE at line 143 etc.

5.       At the preparation of the P1 sample, author should precisely refer to the Odanacatib’s lot used. At line 131 the general term “API” is used, while at line 136 for the preparation of P2 authors refer to sample A.

6.       Authors should provide more details on the role of SSF and MgSt at the size reduction. A reference at studies that have demonstrated the contribution of referred excipients to the particle size reduction would be helpful.

7.       No results should be mentioned in detail at the Materials and methods section (line 122 and 142).

8.       At line 258 it is mentioned that “The use of processing aids led to smaller particles in both A(S) and A(M) than those without using a processing aid”. A possible explanation on this observation should be provided.

9.       The authors should refer to the size span parameter of table 1, since there is no comment about the differences between the values.

10.   Authors should revise the graphical representation of the XRD diffractograms. For example, at figure 2a the XRD pattern of sample A is interrupted. In this vein, line 294 contains a very solid statement without any clear evidence.

11.   The authors claimed that “air entrapment caused the different tableting behaviors, since milled API materials are more cohesive and it is more difficult for air to escape the die during the course of compression”. In this context, please comment on the fact that sample B could not form intact tablets due to lamination, while sample P2.

12.   At line 343 it is mentioned that A2 has significantly lower crystallinity than A1, as suggested by broader peaks in the powder XRD pattern of A2. Further analysis is required for this statement.

13.   The authors should refer to similar studies having been presented in literature and highlight the novelty of the current study.

Author Response

Reviewer 1

The manuscript entitled "Varied Bulk Powder Properties of Micro Sized API within Size Specifications as A Result of Particle Engineering Methods " is a study of the bulk properties’ variability in some different batches of Odanacatib prepared through either jet milling or fast precipitation. The work is interesting as the particle size reduction is one of the several formulation strategies proposed for the improvement of the poorly water-soluble APIs. Although a significant number of experiments was conducted, there are still many concerns that need to be clarified in order for this paper to be considered as acceptable.

  1. The manuscript, and specifically the abstract, needs a thorough check of English language.

We have revised the manuscript, including the abstract, for clarity.

  1. The writing and the logical order of the paper’s Introduction needs to be reorganized. For example, the authors should reduce the size of the paragraph at lines 30-44, since it essentially constitutes a repetition of the same statement. Furthermore, a brief explanation of the “top-down” and “bottom-up” methods should be provided.

We have significantly revised and condensed the introduction section as suggested.

  1. Tabletability is a critical attribute of the examined samples which is evaluated in detail at this study. That being said, the authors should provide the definition of the term and highlight its importance.

We have defined the term “tabletability” and explained its significance.

  1. The definition of the abbreviations when they first appear in the manuscript is mandatory. The authors should thoroughly check the manuscript in order to define all the abbreviations used. For example, BCS at line 42, MTBE at line 143 etc.

We have modified the manuscript as suggested.

  1. At the preparation of the P1 sample, author should precisely refer to the Odanacatib’s lot used. At line 131 the general term “API” is used, while at line 136 for the preparation of P2 authors refer to sample A.

We have modified the text as appropriate.  Sample A was used to prepare both P1 and P2.

  1. Authors should provide more details on the role of SSF and MgSt at the size reduction. A reference at studies that have demonstrated the contribution of referred excipients to the particle size reduction would be helpful.

The processing aids were not sought to more effectively reduce particle size, but rather to reduce disorder (amorphous formation) during jet milling (https://doi.org/10.1002/jps.21998, https://doi.org/10.1016/j.ejps.2015.10.016, https://doi.org/10.1016/j.ijpharm.2013.08.025).  In the authors experience and as other researchers have demonstrated (https://doi.org/10.1016/j.ejps.2021.105782, https://doi.org/10.1021/acs.molpharmaceut.9b00692,) , while jet milling can achieve target particle size, amorphous formation occurs for an appreciable number of development compounds.  This amorphous formation is problematic for phase control and in vivo behavior and can lead to chemical stability challenges.  We sought to explore whether the addition of these processing aids, while reducing disorder, could impact other bulk powder properties including the impact on dissolution kinetics.

  1. No results should be mentioned in detail at the Materials and methods section (line 122 and 142).

Results have been deleted from Lines 122 and 142 as suggested.

  1. At line 258 it is mentioned that “The use of processing aids led to smaller particles in both A(S) and A(M) than those without using a processing aid”. A possible explanation on this observation should be provided.

We have added the following explanation “It is possible that lubricated API particles agglomerated to a less extent and can travel with the jet at a higher speed, leading to a more effective size reduction due to a higher energy of impact. ”

  1. The authors should refer to the size span parameter of table 1, since there is no comment about the differences between the values.

We have added discussion on span.

  1. Authors should revise the graphical representation of the XRD diffractograms. For example, at figure 2a the XRD pattern of sample A is interrupted. In this vein, line 294 contains a very solid statement without any clear evidence.

We feel PXRD data as presented in Figure 2a clearly shows sharper peaks in the unmilled sample A than milled A1, A2, and A3.  We seek more detailed suggestions on how to improve it.

  1. The authors claimed that “air entrapment caused the different tableting behaviors, since milled API materials are more cohesive and it is more difficult for air to escape the die during the course of compression”. In this context, please comment on the fact that sample B could not form intact tablets due to lamination, while sample P2.

Sample B consists of long needles (cotton like), which favors air entrapment.  We added a sentence in the text to explain this.

  1. At line 343 it is mentioned that A2 has significantly lower crystallinity than A1, as suggested by broader peaks in the powder XRD pattern of A2. Further analysis is required for this statement.

We have added the following sentence to compare the two PXRD patterns in support of this statement. “The air entrapment mechanism is also consistent with the fact that it is more difficult for air to escape from the cotton-ball like sample B, which consisted of entangled long needles and laminated upon tablet ejection.”

  1. The authors should refer to similar studies having been presented in literature and highlight the novelty of the current study.

We are unaware of a similar study that looked at different powder properties of API meeting size specifications based on biopharmaceutical criteria.  We appreciate it if the reviewer can point out such a paper that we might have missed.

Reviewer 2 Report

Manuscript is very interesting and original.  The paper was well written. Conclusions consistent with the evidence and arguments presented well good presented. The main question is why the authors did not perform the basic pharmacopoeial test - the dissolution profile study?

Author Response

Reviewer 2

Manuscript is very interesting and original.  The paper was well written. Conclusions consistent with the evidence and arguments presented well good presented. The main question is why the authors did not perform the basic pharmacopoeial test - the dissolution profile study?

We thank the reviewer for the affirmative comment on our work.  We used intrinsic dissolution rate to establish the different dissolution performance of these API lots. We did not perform pharmacopoeial dissolution test because, although certainly useful, it is not critical for reaching that goal. 

Round 2

Reviewer 1 Report

All comments and recommendations were appropriately addressed by the authors. The paper is now compliant with the Pharmaceutics’ standards and, in my opinion, is suitable for publication.